# Advances in Polysaccharide Production Based on the Co-Culture of Microbes

**DOI:** 10.3390/polym15132847

**Published:** 2023-06-28

**Authors:** Wanrong Peng, Xueying Guo, Xinyi Xu, Dan Zou, Hang Zou, Xingyong Yang

**Affiliations:** 1College of Pharmacy, Chengdu University, Chengdu 610106, China; 2College of Life Sciences, Chongqing Normal University, Chongqing 401331, China; 3Antibiotics Research and Re-Evaluation Key Laboratory of Sichuan Province, Chengdu University, Chengdu 610106, China

**Keywords:** co-culture, polysaccharides, microbes, antioxidant, antibacterial, anti-inflammatory

## Abstract

Microbial polysaccharides are natural carbohydrates that can confer adhesion capacity to cells and protect them from harsh environments. Due to their various physiological activities, these macromolecules are widely used in food, medicine, environmental, cosmetic, and textile applications. Microbial co-culture is an important strategy that is used to increase the production of microbial polysaccharides or produce new polysaccharides (structural alterations). This is achieved by exploiting the symbiotic/antagonistic/chemo-sensitive interactions between microbes and stimulating the expression of relevant silent genes. In this article, we review the performance of polysaccharides produced using microbial co-culture in terms of yield, antioxidant activity, and antibacterial, antitumor, and anti-inflammatory properties, in addition to the advantages and application prospects of co-culture. Moreover, the potential for microbial polysaccharides to be used in various applications is discussed.

## 1. Introduction

Polysaccharides, as natural polymers, are widely found in plants, animals, marine organisms, and microorganisms, and their significant biological activity has attracted a lot of attention [1]. Many studies have demonstrated that polysaccharides possess a variety of biological activities, such as antitumor, antioxidant, immunomodulatory, antibacterial, hepatoprotective, and anti-inflammatory [2], which are in turn linked to their molecular weight, monosaccharide composition, glycosidic bonds, and branching degree. However, the low yields of polysaccharides and difficulties in their production and purification have led to limited research on them; this can be improved through co-culture. For example, the co-culture of *Lactobacillus rhamnosus* with *Saccharomyces cerevisiae* increases the yield of polysaccharides by approximately 40%; in addition, co-culture has been shown to increase the levels of metabolites [3].

Microbial co-culture is the process of directly or indirectly cultivating different types of microbial cells in a limited environment (anaerobic or aerobic conditions), simulating the interactions between microorganisms in natural environment [4]. Many biosynthetic genes are usually not expressed or expressed at low levels under laboratory conditions which prevents the realization of the full metabolic potential of these microorganisms [5]. However, co-culture has been proved to be a practical strategy in inducing the biosynthesis of secondary metabolites by activating the silent genes. (Figure 1) [6]. To date, 33 novel secondary metabolites have been successfully obtained from actinomycetes through co-culture with mycolic acid-containing bacteria [7]. Further, the co-culture-induced production of extracellular polymeric substances (EPSs) by cyanobacteria, microalgae, and basidiomycetes was found to be associated with a decrease in the fermentation time and an increase in EPSs production. Co-cultured EPS is a combination of monosaccharides from two purely cultured EPS, resulting in a new structure [8]. Here, we summarized the classification and applications of polysaccharides and co-cultures, compared the EPSs between microbial co-cultures with pure cultures, in addition to discussing the yield of co-cultured EPSs and their effects in antioxidant, antibacterial, antitumor, and anti-inflammatory properties.

## 2. Classification and Application of Polysaccharides

Polysaccharides are classified according to their biological origin as either plant, animal, or microbial polysaccharides. Plant polysaccharides have the advantages of biodegradability, ease of modification, low immunogenicity, and low toxicity [9]. Moreover, they can be conjugated, cross-linked, or functionally modified and then used as nanocarrier materials for polysaccharide delivery systems based on the therapeutic effects of traditional Chinese medicine [9]. Animal polysaccharides can be prepared as nanofibers via electrospinning technology and then applied to tissue engineering, wound healing, and drug delivery [10]. Microbial polysaccharides are carbohydrate polymers with high molecular weights. They are classified as capsular polysaccharides, lipopolysaccharides (LPSs), and EPSs [11]. Microbial EPSs, with shorter production times, easy extraction operations, and the ability to be applied to numerous fields, are becoming popular for research, and these include EPSs produced by endophytic bacteria, extremophiles, microalgae, cyanobacteria, and mixed microbial communities. In food, the EPS of *Lactobacillus plantarum* BR2, with high antioxidant activity and antidiabetic and cholesterol-lowering potential, shows great promise for use in functional foods [12]. From an environmental perspective, EPS produced by *Bacillus* sp. MC3B-22 and *Microbacterium* sp. MC3B-10 (Microbactan), which has biosorption potential when cadmium is involved, offers an alternative approach to heavy metal remediation, especially in industrial waste water [13]. In health care, EPSs produced by *Rhodotorula mucilaginosa* UANL−001L co-cultured with *Escherichia coli* can inhibit biofilm formation and be a promising antimicrobial agent [14]. Further, edible fungal polysaccharides can reach the distal intestine and be assimilated to remodel the gut microbiome (GM) [15]. For example, mushroom polysaccharides improve intestinal inflammation and barrier functions by altering the GM composition and increasing the synthesis of short-chain fatty acids [16]. Moreover, *Ganoderma* polysaccharides can result in remodeling of the disordered GM in rats with type 2 diabetes mellitus and improve host metabolism to achieve antidiabetic effects [17]. The market for microbial polysaccharides is expanding, as they can be produced using industrial waste as a substrate and have various desired properties. This can overcome challenges in production and purification and improve product quality, which could help in the path to their commercialization.

## 3. Microbial Co-Culture

### 3.1. Advantages and Applications of Co-Culture

Microbial co-culture comprises a group of natural microorganisms that is formed by different species or the same species but different strains, in which members can interact with each other symbiotically/antagonistically/chemo-sensitively. This can induce the synthesis of natural products, which can occur via the activation of silent biosynthetic genes through the modulation of signaling molecules and also by providing precursors to promote the biosynthesis of natural products [18,19,20]. Co-culture methods include growth in liquid media, solid−liquid interface systems, membrane separations, spatial separations, and microfluidic systems different co-culture methods are suitable for different microorganisms and objectives [21].

The advantages of co-culture can be maximized by designing co-culture substrates and strains. Co-culture advantages include direct cross-feeding [22] and modularity [23]. The former comprises the conversion of inexpensive, widely available substrate/energy sources into intermediate metabolites, which can subsequently be converted to higher value products through sequential and/or parallel conversion. *S. cerevisiae* and *Scheffersomyces stipitis* are useful strains for ethanol production. Here, the former cannot break down xylose, whereas the latter can absorb both sugars and produce bioethanol efficiently in glucose−xylose mixtures [24]. Modularity reduces the metabolic burdens on each component strain, thus contributing to improved overall bioproduction/biotransformation performance [23]. Different modules of the rosmarinic acid pathway were previously adapted by developing co-cultures of three metabolically engineered *E. coli* strains [25]. The glucose catabolic pathway was disrupted in two strains such that they grew only on xylose, whereas the third strain grew only on glucose. The stability of the co-culture system was improved by reducing competition for carbon sources, and the end product concentration was found to be significantly higher. Co-cultivation also allows for a diverse environment, which might be more suitable for the expression of genes in specific pathways than the homogeneous environment provided by individual strains. This also reduces the adverse effects of byproducts, thus improving biosynthetic performance [23]. Therefore, microbial co-culture has a wide range of application prospects.

### 3.2. Metabolites

The co-culture of microorganisms can be achieved in solid or liquid media and is widely used to study natural interactions and discover new active metabolites [26,27,28]. Co-culture not only resulted in the production of new metabolic products, but also increased the yield of the original metabolites. Five metabolites were isolated after the co-culture of *Streptomyces rochei* MB037 with the fungus *Rhinocladiella similis* 35. Compounds 1 and 2 (Figure 2) were obtained only from co-culture, and compound 1 showed significant antibacterial activity against methicillin-resistant *Staphylococcus aureus*. Compound 3 was present in both co-culture and monoculture, but its yield was significantly increased through co-culture [29]. Six new isoprenylated chromene derivatives and two new isoprenylated phenol glucoside derivatives were produced via the co-culture of *Pestalotiopsis* sp. and *Penicillium bialowiezense,* and compounds 1a and 1b (Figure 2) were found to have potent β-glucuronidase-inhibitory effects [30].

In terms of polysaccharide production through co-culture, it is also possible to obtain new EPSs or increase their yield. When *Cellulomonas* sp. (KYM-7) was co-cultured with *Agrobacterium tumefaciens* (KYM-8), KYM-7 was able to utilize starch but could not produce polysaccharides on its own. Meanwhile, KYM-8 was not able to degrade starch but could utilize the products of KYM-7 to produce polysaccharides. Both can produce only very small amounts of polysaccharides when cultured alone but can produce large amounts of polysaccharides upon co-culture [31]. *Lactobacillus kefiranofaciens* that produces EPSs was cultured in pairs with *L. bulgaricus* and *Streptococcus thermophilus* that do not produce EPSs. Through Fourier-transform infrared spectroscopy, scanning electron microscopy, and monosaccharide composition analysis, the EPS produced by co-culture have different monosaccharide compositions and surface morphologies compared with *L. kefiranofaciens* [32]. Thus, co-culture is a useful strategy to produce natural products with diverse structures and biologically active [33], for which the functions and mechanisms are unknown [20].

## 4. Comparison of Co- and Pure Culture-Produced Polysaccharides

Compared to pure culture, the advantages of microbial polysaccharide production by co-culture outweigh the disadvantages. When co-cultures of *Sanghuangporus lonicericola* and *Cordyceps militaris* were performed, the EPSs content were higher than that in the pure cultures of both *S. lonicericola* and *C. militaris* [34]. Notably, in pure culture, co-culture, and triple fermentation of *Ganoderma lucidum* (A), *Tricholoma matsutake* (B), and *Cordyceps sinensis* (C): the pure culture of A had the highest EPSs content, A-B co-culture had slightly lower EPSs content than A, and A-B-C triple fermentation had the lowest EPSs content. This indicates that not every co-culture can increase the content of EPSs, and the interaction between strains and the selection of medium need to be considered [35]. Co-culture needs to consider the interaction between strains as well as the growth rate, which highlights the advantages of pure culture in comparison. However, this can be refined by pre-experiments and by using pre-and post-inoculation.

Co-culture remained stronger than pure culture in terms of biological activity. The EPSs from the co-culture of *S. lonicericola* and *C. militaris* showed greater scavenging effects on 1,1-diphenyl-2-picrylhydrazyl (DPPH), superoxide anions (O_2_^−^), and hydroxyl radicals (OH∙) than their pure culture counterparts [34]. However, there were some exceptions where co-culture with *Saccharomyces boulardii* was added during the fermentation of *G. lucidum*, and the EPSs that were produced were comparable to those produced by *G. lucidum* monoculture in terms of scavenging 2,2′-azinobis-(3-ethylbenzothiazoline-6-sulfonic acid (ABTS), DPPH, and OH∙ [36]. In the study of lipid peroxidation, the co- and pure cultures of *Marasmius androsaceus* and *C. militaris* were used to extract their EPSs separately and test them with rat liver homogenate; the inhibition rate decreased in the following order: *M. androsaceus* > co-culture > *C. militaris* [37]. This result may be due to the low content of new EPSs produced by co-culture or the new EPSs are inherently less effective in inhibiting lipid peroxidation than *M. androsaceus*. In the three-fungal co-fermentation test, erythrocyte membranes were used as the experimental material, and the lowest relative oxidation rate was determined for three-fungal co-fermentation, followed by two-fungal co-fermentation [35]. Although the experimental materials and evaluation methods were different, the obtained results were similar. Based on suitable strain combination, co-culture polysaccharides had comparable biological activities to those from pure culture, while improving the polysaccharide yield; this could be further improved.

In the isolation, purification, and structural characterization of polysaccharides, at least one fraction can be obtained from pure and co-cultured polysaccharides via ion exchange and gel filtration. One fraction (CP1) was isolated from the co-culture of *G. lucidum* and *Flammulina velutipes*, two fractions (GP1 and GP2-1) were isolated from the monoculture of *G. lucidum*, and one fraction (FP1-1) was isolated from the monoculture of *F. velutipes*. Subsequent infrared spectroscopy revealed that the chemical structure of CP1 was different to that of GP1, GP2-1, and FP1-1 [38]; in addition, four fractions were isolated from the co-culture of *S. lonicericola* and *C. militaris*, and three fractions were isolated from both *S. lonicericola* and *C. militaris* [34]. In summary, alterations in fractions or chemical structure affected the biological activity of polysaccharides. The co-culture technique can interfere with the expression of the secondary metabolite genome through the exchange of signaling molecules, thereby regulating the transcription, translation, and metabolic levels of secondary metabolites, which results in changes in the chemical structure of the secondary metabolites [39]. Therefore, the co-culture method is better at exploiting the potential of microorganisms and promoting the production of microbial polysaccharides than the pure culture method.

## 5. Co-Culture-Derived Microbial Polysaccharides

### 5.1. Polysaccharide Yields

Polysaccharides are highly active and versatile molecules used in the food, chemical, and pharmaceutical industries [40]; improving polysaccharide yields has become essential for the improvement of their commercialization and industrialization [41]. Compared to that with monoculture, co-culture fermentation can markedly improve polysaccharide yields. The production of extracellular proteins and EPSs during co-cultivation of one strain of *Nostoc* sp. (F280) and two strains of *Anabaena cylindrica* (B1611 and F243) was investigated. Co-cultures of F280 and B1611 not only yielded the highest dry weights, but also higher yields of extracellular proteins and cell-bound polysaccharides compared to that of F280 monoculture; meanwhile, F280 and F243 co-culture resulted in the highest content of released polysaccharides [42]. EPS production by *Inonotus obliquus* can also be increased by adding stimulants (VB6, VB1, birchwood, and birch extract), and the addition of different stimulants changes the monosaccharide composition of the polysaccharide, which will affect its physiological activity [43]. Further, a co-culture approach improves polysaccharide production and some physiological activities owing to the production of some novel polysaccharides and secondary metabolites during the co-culture process; the co-culture of different microorganisms might also change the resulting polysaccharide species and structure. In addition to using microbial co-culture to increase the production of polysaccharides, there are also other ways to increase their production (Table 1).

### 5.2. Antioxidant Activity

The antioxidant activity of polysaccharides is closely related to their chemical properties, such as their molecular weight, degree of branching, monosaccharide type, monosaccharide ratio, glycosidic bonds, and modifications [51]. For example, with a decrease in molecular weight, the antioxidant activity of chitosan increases significantly [52]. The basic principle is that molecules with low molecular weights can be incorporated into cells more effectively than those with high molecular weights and donate protons [53]. In an evaluation of polysaccharides extracted from mushroom culture filtrate, the antioxidant activity of polysaccharides was found to depend on the proportion of different combinations of monosaccharides. In simple sugars, rhamnose is a determinant of antioxidant activity. For glycosidic bonds, the side chains arabinose 1→4 and mannose 1→2 are significantly correlated with reducing power, whereas glucose 1→6 and arabinose 1→4 are correlated with DPPH radical scavenging [54]. In terms of polysaccharide modification, hydroxymethylated pumpkin polysaccharide has better O_2_^−^ and OH∙ radical scavenging ability [55]. Garlic polysaccharide was co-heated with ferric chloride to synthesize a garlic polysaccharide–Fe (III) complex, and its O_2_^−^ radical scavenging ability was significantly better than that of garlic polysaccharide, showing a good synergistic effect, but the OH∙ radical scavenging activity was similar [56]. Moreover, the antioxidant capacity of polysaccharides can be improved by heat stress, ultrasound, enzymatic degradation, and sulfonation (Table 2).

#### 5.2.1. DPPH Radical Scavenging Activity

DPPH comprises a stable nitrogen centered on a radical with an unpaired electron, which is responsible for strong absorbance at 517 nm. The amount of this radical is easily reduced upon exposure to proton radical scavengers [61]. DPPH solutions change from dark purple to yellow when antioxidants are present. The co-culture of two strains, *Lactobacillus acidophilus* LA5 and *Bifidobacterium animalis* subsp. lactis BB12, was performed to detect the DPPH free radical scavenging rate using a polysaccharide concentration range of 0–2 mg/mL. With an increase in the concentration, the scavenging rate also increased. Polysaccharides obtained via co-culture also had better DPPH free radical scavenging ability and different heat resistance compared with that of polysaccharides from the LA5 pure culture [62]. However, this ability is not generated by co-culture, as purely cultured EPSs also have the ability to scavenge DPPH and are heat resistant. Co-cultures are aimed at producing EPSs with structural alterations that allow them to improve their DPPH scavenging ability on top of the original one, but not all co-cultures can improve either.

#### 5.2.2. OH∙ Radical Scavenging Activity

A radical is defined as a molecule or molecular fragment that contains one or more unpaired electrons in an atomic or molecular orbital [63]. Free radicals produced through normal biological oxidation are beneficial to the body, as they regulate cell signaling and cytogenesis and inhibit the entry of viruses and bacteria into the body to prevent infection [64]; however, highly reactive OH∙ radicals can react rapidly with biomolecules and can cause serious damage to adjacent organs and tissues [61]. Therefore, it is necessary to improve the antioxidant capacity of polysaccharides. *Photobacterium* sp. LYM-1 and *Aureobasidium pullulans* 2012 were co-cultured and cultured separately; subsequently, the water-soluble polysaccharides (WSPs) produced in fermentations of each strain and the co-culture were extracted and used for OH∙ scavenging assays, separately. With each of the measured concentrations, the WSPs from co-culture exhibited a stronger ability to scavenge OH∙ than the other two WSPs factions [65].

#### 5.2.3. O_2_^−^ Radical Scavenging Activity

Mitochondrial lipid membranes are vulnerable to reactive oxygen species (ROS) attack because mitochondria are the most important source of ROS. O_2_^−^ is considered to be the major ROS because it is formed by adding one electron to molecular oxygen [66]. In a radical scavenging assay, 1,2,3-benzenetriol is rapidly autoxidized in alkaline solutions and produces intermediate products; polysaccharides can interfere with 1,2,3-benzenetriol autoxidation. Therefore, the inhibitory activity of 1,2,3-benzenetriol has been used to assess the ability of polysaccharides to scavenge O_2_^−^ [67]. Co-culture-generated polysaccharides from *Marasmius androsaceus* and *Cordyceps militaris* did not have a good scavenging effect on O_2_^−^ radicals and no quantitative-effect relationship [36]. At the same concentration, the ability of polysaccharides extracted from co-cultures to scavenge O_2_^−^ was not significantly increased compared to that of polysaccharides from monoculture. Notably, the O_2_^−^ scavenging ability of the co-culture-generated polysaccharides was enhanced via chelation with Fe (III). Moreover, *Lepista sordida* and *Pholiota nameko* were co-cultured to produce two polysaccharides (CP-1 and CP-3) and polysaccharides were chelated with iron (III) via –OH and –COOH groups to form stable β-FeOOH structures, CP-1-Fe and CP-3-Fe, which showed antioxidant activities. Furthermore, CP-1-Fe and CP-3-Fe exhibited higher OH∙ and O_2_^−^ radical scavenging ability than CP-1 and CP-3. When the concentration reached 2.4 mg/mL, the O_2_^−^ radical clearance rates of CP-1-Fe and CP-3-Fe could reach 96.41% and 92.83%, which were 108.45% and 159.37% higher than those with the same concentration of CP-1 and CP-3, respectively [68]. Although the co-culture-generated polysaccharides were not effective in O_2_^−^ scavenging, their antioxidant capacity could be further enhanced after forming chelates with metal ions.

### 5.3. Antibacterial Activity

The bacteriostatic mechanism of polysaccharides is mainly through their effect on the cell wall, cell membrane, nucleic acids, proteins, and spores, among other targets, which can result in bactericidal or bacteriostatic effects. The polycationic molecules of chitosan can interact with the anionic groups of the cell wall, thus altering the permeability of the bacterial cell wall [69]. Chitosan can also alter the permeability of the cell membrane, further preventing the entry and exit of nutrients and eventually leading to cell death [70]. However, gram-positive bacteria are resistant to this as they have peptidoglycan walls [71]. Moreover, gram-negative bacteria hydrolyze polysaccharides into monosaccharides that can be utilized as nutrient sources owing to the absence of a peptidoglycan wall [72]. With respect to nucleic acids, which can block the transcription of DNA and interfere with RNA molecules [73], *S. giganteus* H03 polysaccharide exerts antibacterial activity against *S. aureus*, partly because of its ability to bind to plasmid DNA [74]. Proteins are important components of the bacterial structure and are involved in the catalysis of many biochemical reactions and bacterial metabolism [75]; based on this, antimicrobial activity was previously assessed using *Cordyceps* polysaccharides [76]. The decrease in total protein concentrations in *E. coli* cells and the increase in soluble proteins in the cultures indicated a change in membrane permeability, resulting in the inhibition of *E. coli* growth. Moreover, WSPs from cabbage were found to have a good inhibitory effect on spore formation by gram-positive *Bacillus* [77]. In addition to these aspects, polysaccharides can also exert effects on biofilms and intracellular metabolic pathways [78]. These are mechanisms that could lead to their use as antimicrobial agents (Figure 3).

The introduction of quaternary ammonium salts via thiol-Michael addition can impart considerable antibacterial activity to the adducts of polysaccharide for use against *S. aureus* (gram-positive bacteria) and *E. coli* (gram-negative bacteria), but the inhibitory effect on gram-positive bacteria is superior to that on gram-negative bacteria [79]. In addition, the polysaccharides from peony seeds (PSPS) have been modified via sulfation, carboxymethylation, and phosphorylated; the PSPS and their derivatives showed significant antibacterial activity against gram-positive (*Bacillus subtilis* and *S. aureus*) and gram-negative (*E. coli* and *S. typhi*) bacteria. Further, *Enteromorpha prolifera* polysaccharide was formed into degraded polysaccharide (LEP), followed by the synthesis of degraded polysaccharide selenide (Se-LEP). Se-LEP showed stronger inhibitory effects on *E. coli* and weaker inhibitory effects on *S. aureus* than polysaccharide selenide (Se-EP). Se-LEP also showed better inhibitory effects on plant pathogenic fungi [80].

Co- and monocultures were also prepared with *Lepista sordida* and *Pholiota nameko*, and *L. sordida* polysaccharide (LSP), *P. nameko* polysaccharide (PNP), and co-culture-derived polysaccharide (LPP) were extracted; then, the inhibition effect of the polysaccharides in solid and liquid media was determined via the inhibition circle and 96-well plate method. The three polysaccharides were found to have some inhibitory effects on *E. coli*, *S. aureus*, *Listeria monocytogenes*, *Salmonella enteritidis*, and *Lactobacillus* but poorer inhibitory effects on aflatoxin, and the inhibitory effect occurred in the following order: LPP > LSP > PNP. Moreover, co-cultured polysaccharides showed significant inhibitory effects on both gram-positive and gram-negative bacteria. However, the inhibitory effect on mycobacteria was slightly weaker [81]. In contrast, co-culture can be used to obtain not only new metabolites, but also those with good antibacterial activity, which can be a good way to prepare antibacterial agents.

### 5.4. Antitumor Activity

Malignant tumors are currently an important disease that endangers human health [82]; they are mainly treated using surgery, radiotherapy, chemotherapy, targeted therapy, and immunotherapy. However, these therapies can cause physical and economic burdens to patients, whereas polysaccharides not only have antitumor activity, but also are less expensive and have few toxic side effects [83]. They are thus gradually becoming a hot spot for antitumor research. *Trametes versicolor*, fucoidan, and sepia ink polysaccharides can be used as adjuvant therapies for the treatment of tumors [84,85,86]. Polysaccharides mainly exert antitumor effects by blocking the cell cycle or through antitumor angiogenesis, apoptosis-inducing, and immunomodulatory mechanisms (Figure 4) [87], and investigating ways to further improve the antitumor activity of polysaccharides is worthwhile.

The antitumor activity of polysaccharides is influenced by various factors, such as the backbone structure, branched chain properties, and molecular advanced structure [88]. The modification of polysaccharides via sulfation, carboxymethylation, phosphorylation, and acetylation enhances their antitumor activity to some extent [89]. Combining cisplatin with *G. lucidum* polysaccharide not only inhibits tumor growth, but also improves spleen and thymus indices and reduces toxic side effects [90]. Moreover, oridonin in combination with lentinan might enhance the antitumor effect of the latter [91]. Co-culture fermentation can also improve antitumor activity. Using co-culture fermentation with *G. lucidum* and *T. matsutake*, the antitumor activity of co-leavened polysaccharide was found to be superior to that after single bacterial fermentation or with a combination of *G. lucidum* and *T. matsutake* polysaccharides [92]. *T. matsutake* and *C. militaris* were also co-cultured and fermented, and the EPSs of single and mixed bacteria were, respectively, extracted and used to treat B16 melanoma for 72 h. The results showed that the EPSs of mixed bacteria had better antitumor activity than those of single bacteria [93].

### 5.5. Anti-Inflammatory Activity

Inflammation is a physiological response of the organism to protect the host from damage triggered by abiotic and biotic factors, such as bacterial infections and harmful stimuli [94]. Chronic inflammation is closely related to many diseases, such as arthritis, atherosclerosis, and cancer. The treatment of inflammation is focused on steroids and non-steroidal anti-inflammatory drugs [95]; and the mechanisms through which polysaccharides exert anti-inflammatory effects are mainly via the inhibition of inflammation-related cytokines, inhibition of *iNOS* and *COX-2* expression, modulation of NF-κB-related signaling pathways, effects on the immune system, and other factors [96].

A novel polysaccharide (MRP-1) was obtained from *Moringa oleifera*, and treatment with different concentrations of MRP-1 prevented LPS-induced increases in NO and TNF-α production and caused a significant reduction in LPS-induced *iNOS* mRNA expression levels, but had no significant effect on *COX-2* mRNA expression [97]. The anti-inflammatory activity of polysaccharides from *G. lucidum* fermented in co-culture with *Flammulina velutipes* was also assessed based on the inhibition rate of *COX* enzymes (both *COX-1* and *COX-2*), which was 74.5% and 75.4%, respectively, for co-cultured EPS at 3 mg/mL, but only 68.7% and 71.6% for *G. lucidum* polysaccharides and even lower for *F. velutipes* polysaccharides. In addition, co-cultured EPS also exhibited lower cytotoxicity than *G. lucidum* polysaccharides [38].

## 6. Summary and Prospect

Polysaccharides, as ubiquitous biomolecules, are used as food supplements and medicines because they are safe and non-toxic and also have physiological activities, such as antitumor, anti-inflammatory, antibacterial, and antioxidant properties. The generation of new metabolites can be achieved via co-culture, which is a feasible strategy to increase polysaccharide production without compromising their biological activity [82]. However, there are relatively few studies on the use of co-culture to produce polysaccharides, and most of them have only focused on co-culture to improve polysaccharide yields and antioxidant activity, which has limited the development of novel polysaccharides generated through co-culture. Among the aforementioned physiological activities, some mechanisms have been elucidated through in vitro and animal experiments, but more experimental and clinical studies on the practical application of polysaccharides are needed. Together with the continuous optimization of extraction methods and techniques, it is believed that co-culture has obvious significance for promoting the development of polysaccharides.

## Figures and Tables

**Figure 1 polymers-15-02847-f001:**
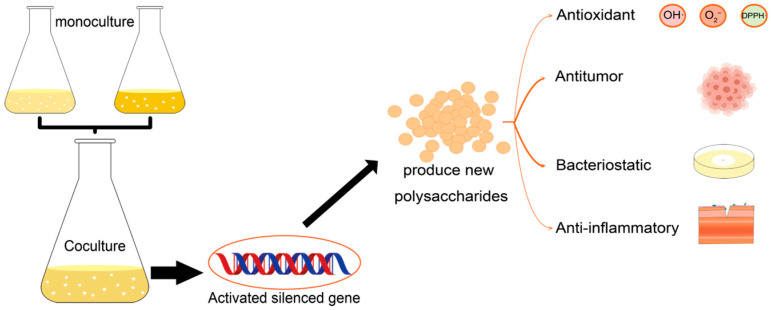
Polysaccharides production by co-culture and its physiological activities.

**Figure 2 polymers-15-02847-f002:**
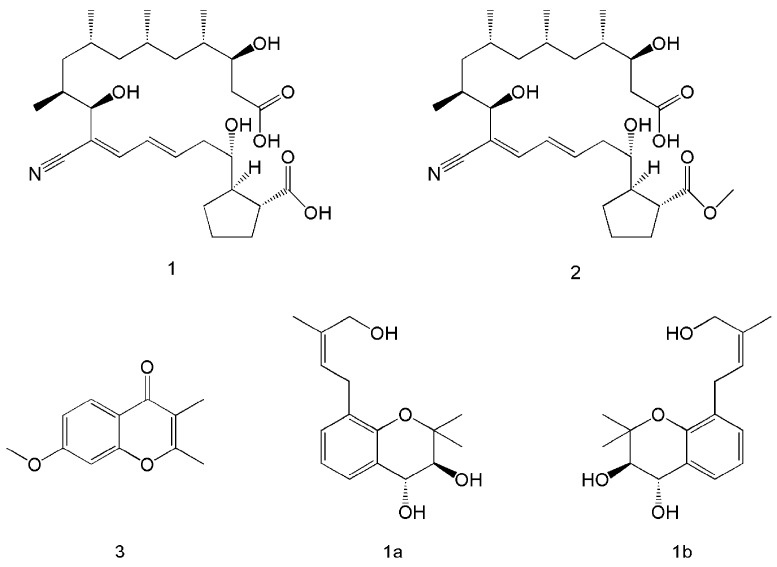
Co-culture to produce metabolites with new structural formulae. Compounds **1**, **2**, and **3** were produced through the co-culture of *Streptomyces rochei* MB037 and *Rhinocladiella similis* 35 [29]. Compounds **1a** and **1b** were produced via the co-culture of *Pestalotiopsis* sp. and *Penicillium bialowiezense* [30].

**Figure 3 polymers-15-02847-f003:**
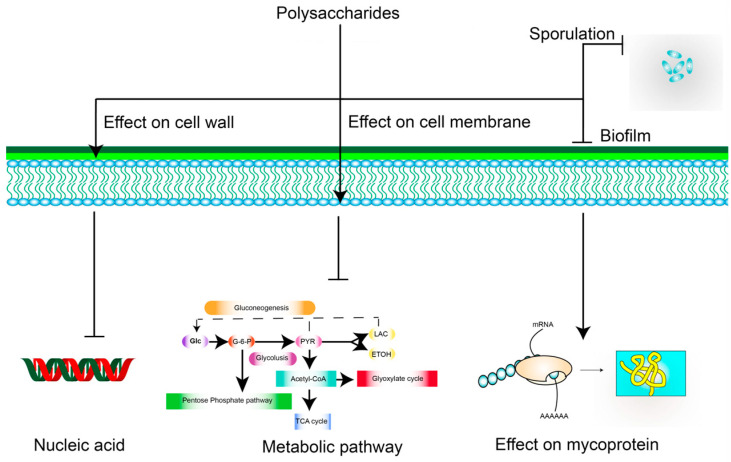
Possible mechanisms underlying the inhibitory effects of polysaccharides against bacteria. Arrows represent activation; ┴ represent inhibition.

**Figure 4 polymers-15-02847-f004:**
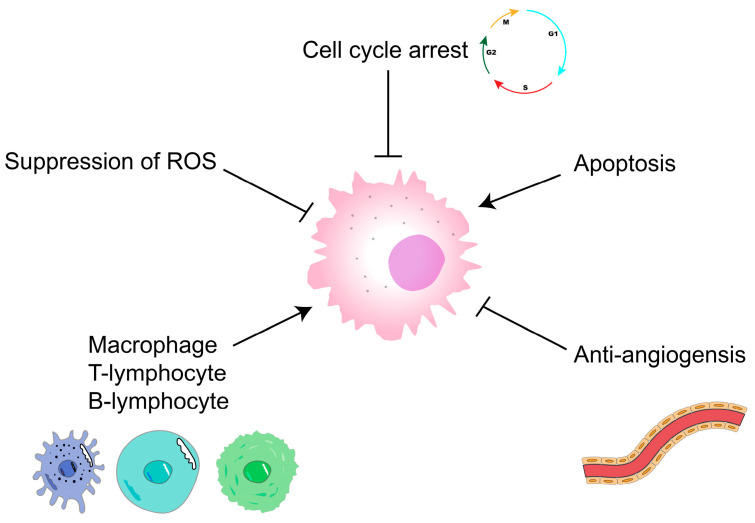
Antitumor mechanisms of polysaccharides. ┴ represent inhibition.

**Table 1 polymers-15-02847-t001:** Methods for increasing polysaccharide yields.

No.	Types of Polysaccharides	Source	Method	Reference
1	Curdlan	*Rhizobium radiobacter*	Add nitrogen source	[44]
2	*Ganoderma lucidum* polysaccharides	*Ganoderma lucidum*	Protoplast mutation	[45]
3	*Cordyceps* polysaccharides	*Cordyceps militaris*	Co-overexpression	[46]
4	*Cyanobacterium Nostoc flagelliforme* EPSs	*Cyanobacterium Nostoc flagelliforme*	Light environment control	[47]
5	*Inonotus obliquus* EPSs	*Inonotus obliquus*	Stimulatory agents	[43]
6	*Paraisaria dubia* polysaccharides	*Paraisaria dubia*	Morphological induction	[48]
7	*Cordyceps militaris* EPSs	*Cordyceps militaris*	Repeated batch approach	[49]
8	*Cordyceps* polysaccharide	*Hirsutella sinensis*	Biosynthetic pathway	[50]

EPSs, extracellular polymeric substances.

**Table 2 polymers-15-02847-t002:** Methods for improving the antioxidant activity of polysaccharides.

No.	Source	Method	Action Targets	Reference
1	*Momordica charantia*	Carboxymethylated	Scavenging O_2_^−^ and OH∙ radicals	[55]
2	*Allium sativum*	Garlic polysaccharide-Fe (III) complex	Scavenging O_2_^−^	[56]
3	*Ganoderma lucidum*	Heat stress	Scavenging OH∙ radicals, DPPH, and ferric reducing antioxidant power	[57]
4	*Phellinus igniarius*	Ultrasound	Scavenging OH∙ radicals ABTS radicals, and ferric reducing ability	[58]
5	*Enteromorpha prolifera*	Enzymatic degradation	Scavenging OH∙, DPPH, and O_2_^−^ radicals	[59]
6	*Pleurotus* and *Streptococcus thermophilus*	Sulfonation	Scavenging OH∙ radicals, ABTS, DPPH, and O_2_^−^	[60]

O_2_^−^, superoxide anions; OH∙, hydroxyl radical; DPPH, 1,1-diphenyl-2-picrylhydrazyl; ABTS, 2,2′-azinobis-(3-ethylbenzothiazoline-6-sulfonic acid.

## Data Availability

Data sharing is not applicable to this article. No data was used for the research described in the article.

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
