# Peer review of "Advances in Polysaccharide Production Based on the Co-Culture of Microbes"

_polymers, 2023, doi:10.3390/polym15132847_

Round 1

Reviewer 1 Report (Previous Reviewer 2)

The authors substantially improved the manuscript. However, there are still minor issues that were not addressed:

Line 10 “with” change to that or which.

Line 44 “exopolysaccharides (EPSs)” was not altered, as well as in Line 215.

The italics in the main text are still missing.

It was substantially improved.

Author Response

  1. Line 10 “with” change to that or which.

Response: Thanks for your suggestion, we have corrected it in MS Line 10.

  1. Line 44 “exopolysaccharides (EPSs)” was not altered, as well as in Line 215.The italics in the main text are still missing.

Response: Thanks for pointing out the mistakes, we have modified them in MS Line 44, 218, 308, and 393.

Reviewer 2 Report (Previous Reviewer 3)

The presentation of this revised manuscript was enough in terms of quality. Moreover, the paper is subjected to following major comments:

In line 148-150: ‘When co-cultures of Sanghuangporus lonic-148 ericola and Cordyceps militaris were performed, the EPSs content were higher than that in the pure cultures of both S. lonicericola and C. militaris’  and In line: 154-155: ‘This shows that EPSs content decreased as the competition became more intense due to an increase in the number of strains’ è contradictory to each other. Need to be clear.

In line 168-169: ‘the inhibition rate decreased in the following order: M. androsaceus > co-culture > C. militaris’ – give the explanation the reason behind either high or low inhibition for pure culture compare to co-culture.

Line 247-249: ‘Polysaccharides obtained via co-culture also had better DPPH free radical scavenging ability and different heat resistance compared with that of polysaccharides from the LA5 pure culture’ – this is general as it is or a function of polysaccharide type or source organism.

Minor editing of English language required

Author Response

  1. 1. In line 148-150: ‘When co-cultures of Sanghuangporus lonicericola and Cordyceps militaris were performed, the EPSs content were higher than that in the pure cultures of both  lonicericola and C. militaris’ and In line: 154-155: ‘This shows that EPSs content decreased as the competition became more intense due to an increase in the number of strains’ è contradictory to each other. Need to be clear.

Response: Thanks for your comment and suggestions. We have revised this sentence in MS Line 155-156.

  1. 2.In line 168-169: ‘the inhibition rate decreased in the following order:  androsaceus > co-culture > C. militaris’ – give the explanation the reason behind either high or low inhibition for pure culture compare to co-culture.

Response: Thank you very much indeed for your comment. When the concentration of EPSs was at 0.1 mg/mL, the inhibition rates of lipid peroxidation by M. androsaceus, co-culture, and C. militaris were 89.4%, 87.1%, and 14.3%, respectively. The EPSs of C. militaris needed to reach 0.8 mg/mL to obtain 91.8% inhibition, the difference in inhibition rates between the two pure cultures was too large. Reason 1: The co-culture produced low levels of new EPSs. Reason 2: The new EPSs produced by the co-culture inhibited lipid peroxidation less than those of M. androsaceus and contained EPSs of M. androsaceus and C. militaris.

  1. 3.Line 247-249: ‘Polysaccharides obtained via co-culture also had better DPPH free radical scavenging ability and different heat resistance compared with that of polysaccharides from the LA5 pure culture’ – this is general as it is or a function of polysaccharide type or source organism.

Response: Thank you very much indeed for your comment. Both LA5 and BB12 have the ability to scavenge DPPH and heat resistance. GC-MS, NMR, FT-IR analysis of pure and co-cultured EPSs revealed that the structure of co-cultured EPSs was different from that of pure EPSs, because the type of polysaccharide was changed by co-culture, which resulted in an increased ability to scavenge DPPH and maintain heat resistance. However, not all of the co-cultured EPSs were able to improve their ability to scavenge DPPH.

This manuscript is a resubmission of an earlier submission. The following is a list of the peer review reports and author responses from that submission.

Round 1

Reviewer 1 Report

Although the topic is very interesting, the topic and content of article did not match. I think the authors should describe the results obtained by various types of Polysaccharide produced by co-culture fermentation. 

Reviewer 2 Report

In this manuscript, the authors performed a comprehensive review on microbe co-culture and its subsequent effect on polysaccharide production, as well as mentioning their possible application. This topic has scientific relevance, all the main issues were addressed, and updated bibliography was used. Therefore, I have only minor considerations that should be addressed before publication:

1.       The English and formatting should be carefully revised throughout the manuscript and some sentences simplified. Some examples of suggested alterations: Line 10: “Microbial polysaccharides are natural carbohydrates that can confer adhesion capacity to cells and protect them from harsh environments. Theses macromolecules are widely used in food…”; Line 74 “mucilaginosa”; Line 106 “Different modules”; Line 145 “Co-culture is thus a useful…”; Lines 154-157 “The production of extracellular proteins and EPS during co-cultivation of one strain of Nostoc sp. (F280) and two strains of Anabaena cylindrica (B1611 and F243) were investigated. Compared to F280 monoculture, F280 and B1611 culture together not only resulted in higher dry weight ….”; Line 193 “4.2.1. DPPH radical scavenging activity”; Lines 211-214 “Photobacterium sp. LYM-1 and Aureobasidium pullulans 2012 were co-cultured and cultured separately and, afterwards, the water-soluble polysaccharides (WSPs) produced in fermentations of each strain and the co-culture were extracted and used for OH∙ scavenging assays, separately.”; Line 236 “than CP-1 and CP-3.”

2. The terminology of EPS should be clarified, since it appears as exopolysaccharides and extracellular polysaccharides. Only one term should be applied, and I suggest extracellular polymeric substances.

3.  Table 1 again the different terms regarding the same (exopolysaccharides and extracellular polysaccharides). In addition, only in Nostoc flagelliforme was added the type of microorganism (cyanobacterium). It could be useful to add it in all examples.

4.       Table 2 since all examples are polysaccharides and it is already stated in the headline, I would remove it from all the examples to simplify the reading.

5.       Italics are missing in Line 261 “Bacillus”, Line 348 “in vitro” as well as in the References section.

Reviewer 3 Report

The presentation of this manuscript was not good enough in terms of quality and methodology. Moreover, the paper is subjected to following major comments:

1.      I am doubt on effect of cell concentration or cell growth rate.

2.      Importance and aim of this study are not clear.

3.      There are no creative or comparative analysis of Polysaccharide production, their properties, activities even no comparison between pure and Co-Culture of Microbes.

4.      Overall, there are lack of sufficient novelty and interest of this manuscript.